# Development of a Set of Assessment Tools for Health Professionals to Design a Tailored Rehabilitation Exercise and Sports Program for People with Stroke in South Korea: A Delphi Study

**DOI:** 10.3390/healthcare11233031

**Published:** 2023-11-24

**Authors:** Minyoung Lee, Yoon Park, Seon-Deok Eun, Seung Hee Ho

**Affiliations:** 1Department of Holistic Integrative Healing Studies, Seoul Cyber University, Seoul 01133, Republic of Korea; wharen88@gmail.com; 2Department of Clinical Rehabilitation Research, National Rehabilitation Research Institute, Seoul 01022, Republic of Korea; ban3@korea.kr; 3Assistive Technology Research Team for Independent Living, National Rehabilitation Research Institute, Seoul 01022, Republic of Korea; esd7786@korea.kr; 4Department of Healthcare and Public Health Research, National Rehabilitation Research Institute, Seoul 01022, Republic of Korea

**Keywords:** Delphi survey, rehabilitation exercise, physical function, mental function, social ability, people with stroke

## Abstract

We developed a set of assessment tools for health professionals to evaluate the physical functions, mental functions, and social abilities of people with stroke (PWS) from 6 months to 3 years after stroke onset, to design a tailored “Rehabilitation Exercise and Sports” (RES) program, which the South Korean government was required to provide by the Act on Guarantee of Right to Health and Access to Health Services for people with disabilities. Since previous studies mainly dealt with the chronic stage of PWS, it would not be appropriate to apply assessment tools used in previous studies, as they are not compatible with the time window (6 months to 3 years) used to define the target population of the RES program. We reviewed the literature to identify evaluation factors and assessment tools applied in previous studies, and developed a Delphi questionnaire with closed-ended questions based on the literature review’s results and supplementary open-ended questions. A 20-expert panel conducted four rounds of the Delphi survey, including two rounds to determine evaluation factors and two rounds to determine assessment tools. The Delphi survey revealed that 22 evaluation factors and 24 corresponding assessment tools reached consensus among the experts. However, no assessment tools reached consensus for three evaluation factors: muscle endurance, flexibility, and dynamic balance. A comprehensive set of assessment tools would be useful for health professionals to understand the health status of PWS from 6 months to 3 years after stroke onset, and help the design of tailored RES programs.

## 1. Introduction

South Korea is expected to face an ultra-aged society, in which more than 20% of the total population will be 65 years or older by 2025. Along with this high aging rate, the population of patients with stroke (PWS) is also growing, causing medical costs to be approximately six times higher than those of the general population [1]. Stroke also represents the third most prevalent type of disability [2,3]. Furthermore, more than 25% of people with a history of stroke are at an increased risk of recurrence [4]. Stroke recurrence is influenced by a number of metabolic risk factors, such as hypertension, impaired glucose control, dyslipidemia, obesity, and low cardiorespiratory fitness [5,6].

Exercise is a safe, inexpensive, and effective method for reducing metabolic risk factors in PWS with minimal side effects [7,8]. Moreover, exercise improves not only physical function, such as strength, endurance, balance, and walking performance [8,9,10,11,12,13], but also psychological functions in PWS [14]. However, the participation rate of non-disabled people in exercise is 60.8%, whereas the rate for people with disabilities (PWD) is only 24.2% [15]. Furthermore, well-equipped infrastructure and services for rehabilitation or exercise in the community is lacking, eventually forcing patients to be readmitted to another hospital or to be left alone at home after discharge [16].

In South Korea, to address the issue of health inequality between people with and without disabilities, the Act on Guarantee of Right to Health and Access to Health Services for PWD (PWD Health Rights Act) has been in effect since 2017 [17]. In particular, Article 15 stipulates that the government should strive to improve the physical and mental functions, and social abilities of PWD or those expected to become PWD within a certain period of time after an injury or disease by providing them with “Rehabilitation Excise and Sports” (RES) programs [17]. “A certain period of time” is usually interpreted as approximately between 6 months and 3 years after disease or injury onset. Therefore, this regulation aims to help patients discharged from acute hospitals settle into society early and encourage them to enjoy sports upon their return.

As suggested by its name, RES is an intermediate step between “therapeutic exercise” provided in hospitals and “sports for all” performed in the community. It should be possible to simultaneously achieve the goals of “recovery of function” and “enhancement of physical fitness” for PWD. Furthermore, the improvement of mental function and social ability is also included in the purpose of the RES program, as stipulated in the PWD Health Rights Act. PWS are expected to have wide variations in physical function levels, depending on the stroke type and severity. There are also differences in mental function and social ability, depending on individual personalities and the socioecological environment to which each person belongs. Therefore, to achieve the goals of the RES program for PWS, it is important to design interventions tailored to each individual or groups of individuals with similar types, taking into account the physical, mental, and social abilities of each subject based on the biopsychosocial approach. A prerequisite for adopting this approach is to accurately understand the health status of PWS during this period and provide health professionals with the necessary information to design tailored interventions.

Many studies have reported the effects of exercise on community-dwelling PWS. These studies ranged from aiming to improve physical functions, such as muscle strength [18,19,20,21,22,23,24,25,26,27,28,29,30], balance [18,19,21,25,27,28,31,32,33,34,35,36,37,38], walking [20,21,26,27,28,31,32,33,34,35,36,37], cardiopulmonary endurance [20,21,22,23,28,30,31,38,39], upper/lower extremity function [22,26,34,40,41,42], flexibility [19,22,31], and fall prevention [19,31,32], to enhancing cognition [23,28], depression [32,43], and quality of life [20,24,26,33,36,44]. In these studies, various evaluations were applied using specific assessment tools to confirm the health status of PWS and the intervention effects. However, since these studies mainly dealt with the chronic stage of PWS for more than 3 years, it would not be appropriate or sufficient to apply the assessment tools used in previous studies, as they are not compatible with the time window (6 months to 3 years) used to define the target population of the RES program. Moreover, as the RES program aims to improve overall health, including physical and mental functions, and social abilities, it is necessary to develop a comprehensive assessment toolset that can evaluate the overall health status beyond specific areas.

Therefore, this study aims to develop a set of assessment tools necessary for confirming the physical and mental functions, and social abilities of PWS from 6 months to 3 years after stroke onset, constituting the target population of the RES program. In particular, we used the Delphi method to collect expert opinions on the development of this set of assessment tools. 

## 2. Methods

### 2.1. Study Design

This study adopted the Delphi method to develop a set of assessment tools to help health professionals evaluate the health status of PWS and design a tailored RES program. The Delphi method is usually anonymous, iterative, and survey-based. This method is designed to transform opinions into a group consensus, with each person having an equal voice in the decision-making process [45]. This method has been used to enhance effective decision-making in health and social care [45].

### 2.2. Development of a Set of Assessment Tools

The hierarchical structure for evaluation factors and a set of assessment tools corresponding to each evaluation factor were developed through the following four steps (Figure 1).

#### 2.2.1. Step 1. Literature Review

First, the researchers searched for articles published in the last 10 years, from 2011 to the present, which included the combination of the words “Exercise or Sports or Muscle strength or Muscle endurance or Cardiovascular endurance or Flexibility or Body composition or Balance or Coordination or Range of motion or Upper/Lower extremity function or Activity of daily living or Cognition or Depression or Fear of falling or Quality of life” and “Stroke” in the article title, on Scopus, CINAHL, and Web of Science. A total of 98 articles were identified. Second, among the identified articles, only 26 studies [18,19,20,21,22,23,24,25,26,27,28,29,30,31,32,33,34,35,36,37,38,39,40,41,42,43,44] that applied exercise or sports interventions to PWS for a period of time, on an individual- or group-basis in community settings were selected. The researchers tried to identify the functions targeted for improvement by the authors by applying exercise or sports (e.g., muscle strength and flexibility), and also tried to identify the tools applied to assess these functions before and/or after intervention. In the current study, our research team defined the former as an “evaluation factor” and the latter as an “assessment tool”. In this step, the researchers identified 14 evaluation factors and 58 assessment tools among the 26 selected studies. Third, among the identified assessment tools, the researchers determined a total of 47 tools whose reliability or validity had been verified in previous studies. The researchers developed a draft version of Delphi items based on the determined 14 evaluation factors and 47 evaluation tools.

#### 2.2.2. Step 2. Building a Draft Version of the Hierarchical Structure for Evaluation Factors and Determining Corresponding Assessment Tools for Each Evaluation Factor

Before the initiation of the Delphi process, an expert steering committee was convened to build a draft version of a hierarchical structure for evaluation factors and determine corresponding assessment tools for each evaluation factor. This step was intended to systematically classify the evaluation factors and assessment tools by grouping factors and tools with similar purposes. Steering committee members were selected from those who met one of the following criteria: (1) >10 years of exercise intervention experience for community-dwelling PWS, or (2) >10 years of teaching experience as a professor in related fields such as rehabilitation medicine, sports medicine, special physical education, kinesiology, and physical therapy. First, a steering committee of 10 experts listed the evaluation factors and assessment tools as broadly and inclusively as possible to ensure the generation of a comprehensive list based on possible evidence from the reviewed literature. Second, experts suggested other evaluation factors or assessment tools in addition to those reviewed, based on their expert knowledge and experience. Third, experts built a draft version of a hierarchical structure by grouping factors with similar purposes and corresponding assessment tools for each evaluation factor using the design-thinking process. The design-thinking process provided the optimal solution derived from the opinions of steering committee members through repeated deductive and conductive verification [46,47].

#### 2.2.3. Step 3. Determination of Evaluation Factors

The researchers developed a questionnaire in which each evaluation factor consisted of a Delphi item, and these items were presented based on the draft version of the hierarchical structure discussed in Step 2. The questionnaire was based on a 5-point Likert scale ranging from 1 to 5 (1 = strongly disagree, 2 = disagree, 3 = neither agree nor disagree, 4 = agree, and 5 = strongly agree). A free-text comment box for each item was included in the survey for the expert panel to suggest the addition, removal, or modification of items. A web-based platform was employed to gather input from the expert panel to maintain anonymity and avoid individual bias in consensus formation during the Delphi survey. In the current study, a Delphi survey was requested by an expert panel in two rounds [20] between September 2022 and October 2022. After completing Delphi Round 1, all text comments on the questionnaire were evaluated, modifications were made to reduce ambiguity if necessary, and new elements were added if deemed appropriate. In Delphi Round 2, all ratings from the first round were analyzed and expressed as the percentage of respondents with an item score ranging from 1 to 5 for each element. The experts who responded in the first Delphi round subsequently received the summarized results and were requested to reconsider their ratings. The final version of the assessment factors was determined based on consensus from the second round.

#### 2.2.4. Step 4. Determination of Assessment Tools

The researchers developed a questionnaire in which each assessment tool consisted of a Delphi item. The Delphi items (i.e., assessment tools) were classified and presented corresponding to each evaluation factor. In each Delphi item, “importance” and “feasibility” of the assessment tool were rated based on a 5-point Likert scale, ranging from 1 to 5 (1 = strongly disagree, 2 = disagree, 3 = neither agree nor disagree, 4 = agree, and 5 = strongly agree). The Delphi survey for assessment tools was conducted in two rounds, between November 2022 and December 2021. The detailed Delphi survey process was similar to the one used to determine evaluation factors described in Step 3.

### 2.3. Expert Panel Selection

Purposive and snowballing sampling strategies were employed to recruit experts among those who met one of the following criteria [48]: (1) >10 years of exercise intervention experience for community-dwelling PWS, or (2) >10 years of teaching experience as a professor in related fields, such as rehabilitation medicine, sports medicine, special physical education, kinesiology, and physical therapy. This study was premised on the stage after a doctor comprehensively evaluated the patient’s disease and screened whether he or she could perform exercise considering risk factors, focusing on determining a set of assessment tools necessary to design an intervention tailored to each individual or groups of individuals with similar types. Thus, doctors were not included as a panel in the current study. A total of 20 multidisciplinary experts specializing in rehabilitation medicine (n = 3), sports medicine (n = 2), kinesiology (n = 8), special physical education (n = 4), and physical therapy (n = 3) were included. Among them, 17 experts (69%) had doctoral degrees, and the other experts had master’s degrees (19%). The mean work experience in the related field was 23.45 ± 5.9 years. The response rates for the determination of assessment factors and assessment tools in rounds 1 and 2 were 100% (n = 20) and 95% (n = 19), respectively (Table 1).

### 2.4. Data Analysis

Analysis of responses to closed-ended questions of the Delphi survey for evaluation factors (Rounds 1 and 2) and tools (Rounds 1 and 2) was performed after the completion of each round using SPSS 21. Results of descriptive statistics were presented as mean and standard deviation. Content validity was verified using the content validity of individual items (I-CVI) and the positive coefficient of variation (CV), a relative measure of statistical dispersion. I-CVI was calculated as the number of experts giving a rating of 4 and 5 divided by the total number of experts [22,40]. We used a mean ≥ 4.0, I-CVI ≥ 0.75 (75%), and CV < 0.50 [18,22] as the stop criteria. In this approach, the opinions of most experts are agreed upon if the mean, I-CVI, and CV values meet the stop criteria [18,22]. During the analysis of the results of Delphi survey for the determination of assessment tools, opinions of most experts were considered “agreed upon” only when the stop criteria were satisfied in both “importance” and “feasibility” aspects.

## 3. Results

### 3.1. Development of a Hierarchical Structure for Evaluation Factors

The research team identified the function that the authors intended to improve by applying the intervention or confirming the degree of improvement through evaluation in community settings in previous studies, considering it as the evaluation factor. A total of 14 evaluation factors and 47 assessment tools were determined among 26 selected studies. The factors identified through the literature review were as follows: muscle strength and endurance [18,19,20,21,22,23,24,25,26,27,28,29,30,37], cardiovascular endurance [20,21,22,23,28,30,31,38,39], flexibility [19,22,31], body composition [27], balance [18,19,21,25,27,28,31,32,33,34,35,36,37,38], coordination [40], range of motion [22], upper and lower extremity function [22,26,34,36,40,42], activities of daily living (ADL) [42], cognition [23,28], depression [32,43], fear of falling [32], and quality of life [20,24,26,33,36,44]. In addition to the evaluation factors identified in previous studies, the steering committee members added several factors to be evaluated before and/or after implementing the RES program for PWS in the design-thinking process. Added evaluation factors were instrumental activities of daily living (IADL), self-efficacy, body awareness, disability acceptance, communication skills, and social participation. Thus, a total of 20 evaluation factors were determined, and 6 assessment tools to evaluate this were suggested additionally.

The research team and steering committee members classified the evaluation factors based on several criteria (Figure 2). First, it was largely categorized into the following three factors, defined as the purpose of the RES program in the PWD Health Rights Act [17]: (1) physical function (n = 11), including muscle strength, muscle endurance, cardiovascular endurance, flexibility, body composition, balance, coordination, range of motion, upper/lower extremity, ADL, and IADL; (2) mental function (n = 5), including cognition, depression, fear of falling, self-efficacy, and body awareness; and (3) social ability (n = 4), including disability acceptance, communication skills, social participation, and quality of life.

Second, physical function was subcategorized into (1) health-related factors (n = 5), (2) skill-related factors (n = 2), and (3) ADL-related factors (n = 4) (Figure 2). According to previously reported guidelines [49], physical fitness is classified into health-related and skill-related factors. Health-related factors consist of components of physical fitness that are related to good health, commonly defined as body composition, cardiovascular endurance, flexibility, muscular endurance, and strength. Skill-related factors consist of physical fitness components that have a relationship with enhanced performance in sports and motor skills, commonly defined as agility, balance, coordination, power, speed, and reaction time. However, among these factors, agility, power, speed, and reaction time were considered too difficult to apply to subjects of the RES program by the steering committee members. However, since factors related to physical fineness are not sufficient to accurately evaluate the health status of PWS, the steering committee members additionally suggested ADL-related factors to evaluate the disability-specific function of PWS. As a result, evaluation factors related to physical function were classified as detailed in Figure 2.

### 3.2. Delphi Survey for the Determination of Evaluation Factors

A Delphi survey was conducted with 20 Delphi items determined through literature review and design thinking.

Round 1 of the Delphi survey revealed that the expert panel agreed that all 20 evaluation factors were required for PWS to perform the RES program, satisfying the stop criteria (mean ≥ 4.0, I-CVI ≥ 0.75, and CV < 0.50). In Round 1, several experts suggested in a free text comment box that “muscle tone” needed to be added to ADL-related factors, and that balance needed to be divided into “static balance” and “dynamic balance”, considering the diversity of functional levels according to the severity of disability. Therefore, in Round 2, a total of 22 items were generated to reflect these opinions, and the mean, I-CVI, and CV values of all Delphi items satisfied the stop criteria, indicating that the expert panel reached consensus on all 22 evaluation factors (Table 2).

### 3.3. Delphi Survey for the Determination of Assessment Tools

A Delphi survey with 53 items determined through literature review (47 items) and the design-thinking process (6 items) was conducted in Round 1. After completing Round 1, there was an opinion to add an assessment tool for muscle tone through a free text comment box; accordingly, in Round 2, a Delphi was conducted on a total of 54 Delphi items. Round 2 of the Delphi survey revealed that the expert panel agreed that 24 assessment tools among the 54 Delphi items were important and feasible for the evaluation of the health status of PWS during their performance of RES programs in community settings, meeting the stop criteria (mean ≥ 4.0, I-CVI ≥ 0.75, and CV < 0.50) (Table 3).

For assessment tools to evaluate health-related factors in physical function, manual muscle tests (importance: mean = 4.42, I-CVI = 0.85, CV = 0.15; feasibility: mean = 4.26, I-CVI = 0.85, CV = 0.15), forced vital capacity (FVC) and forced expiratory volume in one second (importance: mean = 4.05, I-CVI = 0.85, CV = 0.13; feasibility: mean = 4.0, I-CVI = 0.85, CV = 0.11), and body composition analysis (importance: mean = 4.11, I-CVI = 0.85, CV = 0.13; feasibility: mean = 4.21, I-CVI = 0.85, CV = 0.15) were considered appropriate.

For assessment tools to evaluate skill-related factors in physical function, functional reach tests (importance: mean = 4.37, I-CVI = 0.9, CV = 0.14; feasibility: mean = 4.32, I-CVI = 0.9, CV = 0.13), timed up and go (importance: mean = 4.37, I-CVI = 0.75, CV = 0.19; feasibility: mean = 4.26, I-CVI = 0.75, CV = 0.18), Berg balance scale (importance: mean = 4.32, I-CVI = 0.85, CV = 0.15; feasibility: mean = 4.21, I-CVI = 0.8, CV = 0.16), trunk impairment scale (importance: mean = 4.21, I-CVI = 0.9, CV = 0.12; feasibility: mean = 4.05, I-CVI = 0.85, CV = 0.13), and postural assessment scale for stroke (importance: mean = 4.21, I-CVI = 0.8, CV = 0.16; feasibility: mean = 4.11, I-CVI = 0.85, CV = 0.13) were considered appropriate.

For assessment tools to evaluate ADL-related factors in physical function, goniometer (importance: mean = 4.58, I-CVI = 0.95, CV = 0.11; feasibility: mean = 4.47, I-CVI = 0.95, CV = 0.11), motor assessment scales (importance: mean = 4.42, I-CVI = 0.85, CV = 0.15; feasibility: mean = 4.32, I-CVI = 0.85, CV = 0.15), manual function test (importance: mean = 4.26, I-CVI = 0.9, CV = 0.13; feasibility: mean = 4.21, I-CVI = 0.9, CV = 0.12), modified Ashworth scale (importance: mean = 4.53, I-CVI = 0.95, CV = 0.11; feasibility: mean = 4.21, I-CVI = 0.9, CV = 0.16), modified Bathel index (importance: mean = 4.32, I-CVI = 0.9, CV = 0.13; feasibility: mean = 4.00, I-CVI = 0.8, CV = 0.18), and Korean IADL (importance: mean = 4.37, I-CVI = 0.95, CV = 0.11; feasibility: mean = 4.16, I-CVI = 0.9, CV = 0.12) were considered appropriate.

For assessment tools to evaluate mental function, mini-mental status examinations (importance: mean = 4.47, I-CVI = 0.95, CV = 0.11; feasibility: mean = 4.32, I-CVI = 0.9, CV = 0.13), patient health questionnaires (importance: mean = 4.32, I-CVI = 0.85, CV = 0.15; feasibility: mean = 4.11, I-CVI = 0.85, CV = 0.13), self-efficacy (importance: mean = 4.53, I-CVI = 0.95, CV = 0.11; feasibility: mean = 4.26, I-CVI = 0.95, CV = 0.10), fear of falling questionnaires (importance: mean = 4.42, I-CVI = 0.9, CV = 0.13; feasibility: mean = 4.26, I-CVI = 0.9, CV = 0.13), and body awareness questionnaires (importance: mean = 4.42, I-CVI = 0.95, CV = 0.11; feasibility: mean = 4.05, I-CVI = 0.9, CV = 0.15) were considered appropriate.

For assessment tools to evaluate social ability, disability acceptance (importance: mean = 4.58, I-CVI = 0.95, CV = 0.11; feasibility: mean = 4.32, I-CVI = 0.95, CV = 0.11), Holden communication scale (importance: mean = 4.47, I-CVI = 0.9, CV = 0.13; feasibility: mean = 4.26, I-CVI = 0.9, CV = 0.13), participation in social activities and environmental factors (importance: mean = 4.47, I-CVI = 0.95, CV = 0.11; feasibility: mean = 4.26, I-CVI = 0.95, CV = 0.10), European quality-of-life-5-dimensions questionnaires (EQ-5D, importance: mean = 4.26, I-CVI = 0.75, CV = 0.18; feasibility: mean = 4.26, I-CVI = 0.85, CV = 0.15), and the World Health Organization (WHO) quality of life (importance: mean = 4.42, I-CVI = 0.95, CV = 0.11; feasibility: mean = 4.21, I-CVI = 0.95, CV = 0.10) were considered appropriate.

Among the evaluation factors, there was no assessment tool that experts agreed to be appropriate for the evaluation of muscle endurance, flexibility, and dynamic balance. In other words, while experts recognized that these functions should be evaluated, the assessment tools presented in this Delphi survey were not appropriate for the target population of the RES program.

In Rounds 1 and 2, several experts suggested in a free-text comment box that assessment tools needed to be different, depending on the level of severity of stroke or the ability to walk; step or walking tests were deemed not appropriate due to the risk of falling, some tools would be difficult to apply in community settings because they take a long time to evaluate, and self-report tests would be difficult to apply to those with cognitive impairment or hand function problems.

## 4. Discussion

In this study, we developed a set of assessment tools for health professionals to evaluate the physical, mental, and social abilities of PWS from 6 months to 3 years after stroke onset. As such, the health status of PWS during this period could be accurately understood and health professionals could be provided with the information necessary to design a tailored RES program. By developing a hierarchical structure for the evaluation factors, we examined whether the factors evaluating overall health status were included without omission. Through the Delphi survey, expert consensus was reached on a total of 22 evaluation factors and 24 assessment tools. However, experts judged that none of the evaluation tools presented in the Delphi items were appropriate for assessing muscle endurance, flexibility, and dynamic balance.

To develop the hierarchical structure, researchers and experts derived 22 evaluation factors based on several criteria. First, it was largely categorized into three areas: physical function, mental function, and social ability, as defined by the purpose of the RES program in the PWD Health Rights Act of South Korea [17]. The purpose of the RES program is in line with the definition of health presented by the International Classification of Functioning, Disability, and Health (ICF) [50]. The ICF consists of four areas: body and structure, activity and participation, environment, and individual factors, each of which includes several domains that correspond to the evaluation factors of this study. Since the ICF consists of a total of 362 second-level categories and the maximum number of factors for health examination, it is difficult to apply these criteria to all situations. Therefore, an ICF Coreset composed of only the categories appropriate for specific diseases and onset periods has been developed for several diseases. For neurological conditions, Coresets for acute, post-acute, and sub-acute phases have been developed and published [51]. However, the Coreset for the post-acute or sub-acute phase is composed of a total of 116 second-level categories, making it difficult to apply to chronic patients in community settings. Therefore, the evaluation factors derived from the results of this study could serve as a reference for an ICF domain to be applied to community-based exercise in the chronic stage.

Second, physical functions were subcategorized into health-related, skill-related, and ADL-related factors based on previously reported guidelines [49]. In particular, the inclusion of ADL-related function was based on the fact that the RES program aims to improve recovery of function as well as physical fitness in PWS within 6 months to 3 years of discharge, differentiating it from sports programs for all PWS (with an emphasis on physical fitness). Therefore, this systematic structure can play a role in guiding the design of interventions to meet the purpose of the RES defined by the PWD Health Rights Act.

Through the Delphi survey, expert opinions were reached on 22 evaluation factors and 24 assessment tools. However, it is necessary to pay attention to the expert opinion that none of the presented assessment tools were deemed appropriate as evaluation factors for muscle endurance, flexibility, and dynamic balance, which are the main assessment tools that require dynamic motion. This finding suggests that assessment tools such as (1) Step test, Step-box test, 30 s chair stand test for measuring muscle endurance, (2) Sit-and-reach test, Trunk-forward-flexion test, Shoulder-flexibility test for measuring flexibility, and (3) Timed up and go, Berg balance scale, Tinetti test, Short-physical-performance battery, 4-square step test for measuring dynamic balance are not appropriate for PWS from 6 months to 3 years after stroke onset, although some of these tools were confirmed to be reliable or valid for PWS in acute or chronic stages in previous studies [52,53,54]. Therefore, it is necessary to develop new assessment tools suitable for PWS at this age. In addition, it is necessary to refer to the opinions presented by experts in the open-ended questionnaire to develop new evaluation tools to evaluate muscle endurance, flexibility, and dynamic balance for PWS during this period, including (1) adjusting assessment tools based on the level of severity of the stroke or the ability to walk, and (2) recognizing that step or walking tests were not appropriate due to the risk of falling.

There are several limitations in the current study. Firstly, the expert panel for the determination of evaluation factors and assessment tools consisted only of professors and instructors; therefore, the opinions of PWS were lacking. In follow-up studies, assessment tools should be supplemented to reflect the opinions of PWS. Secondly, the reliability and validity of each assessment tool for PWS from 6 months to 3 years after stroke onset have not been verified. Follow-up studies should analyze the reliability and validity of the evaluation tools selected through this study.

## 5. Conclusions

A set of assessment tools for health professionals to design a tailored “Rehabilitation Exercise and Sports” program for PWS from 6 months to 3 years after stroke onset was developed using the Delphi survey. The survey revealed that 22 evaluation factors and 24 corresponding assessment tools reached consensus among experts, although no assessment tools reached consensus for three evaluation factors: muscle endurance, flexibility, and dynamic balance, suggesting the necessity of developing new assessment tools for these evaluation factors. Health professionals would thus be able to gain a more accurate understanding of the health status of PWS during this time, as well as be provided with the information needed to design a tailored RES program.

## Figures and Tables

**Figure 1 healthcare-11-03031-f001:**
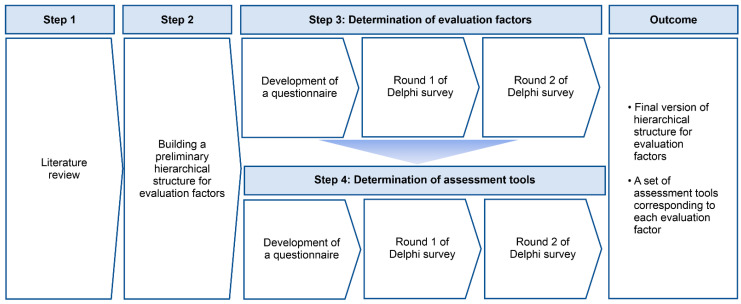
Development of a set of assessment tools through the Delphi method.

**Figure 2 healthcare-11-03031-f002:**
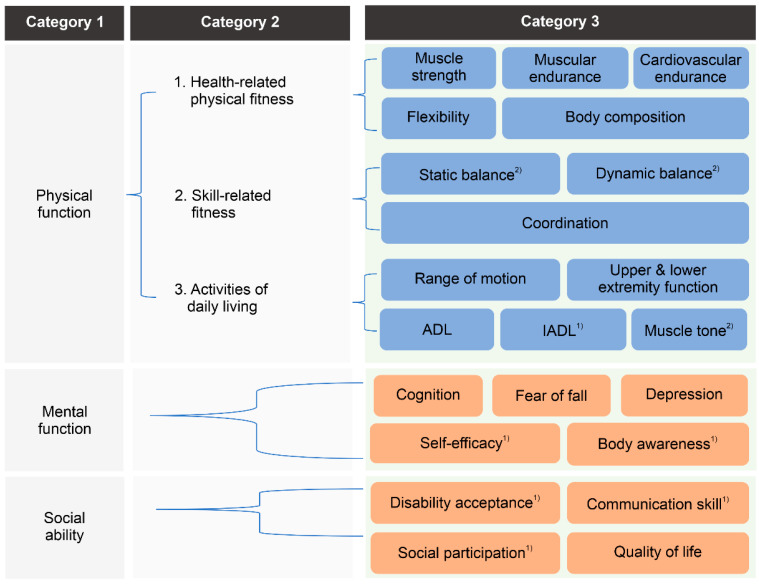
Development of a hierarchical structure for evaluation factors. ADL, activities of daily living; IADL, instrumental activities of daily living. ^(1)^ Added by the steering committee during the design-thinking process. ^(2)^ Added to reflect the results of the first Delphi survey for the determination of evaluation factors.

**Table 1 healthcare-11-03031-t001:** The composition of the expert panel who participated in the Delphi survey.

n	Major	Position	ID	Degree	Major Field Experience (Year)
1	Rehabilitation Medicine	Professor	W1	MD	27
2	Rehabilitation Medicine	Professor	W2	MD	24
3	Rehabilitation Medicine	Professor	W3	MD	16
4	Sports Medicine	Professor	W4	PhD	25
5	Sports Medicine	Instructor	W5	MS	18
6	Kinesiology	Professor	W6	PhD	30
7	Kinesiology	Professor	W7	PhD	23
8	Kinesiology	Professor	W8	PhD	18
9	Kinesiology	Professor	W9	PhD	25
10	Kinesiology	Professor	W10	PhD	15
11	Kinesiology	Professor	W11	PhD	27
12	Kinesiology	Instructor	W12	MS	25
13	Kinesiology	Instructor	W13	MS	24
14	Special Physical Education	Professor	W14	PhD	32
15	Special Physical Education	Professor	W15	PhD	20
16	Special Physical Education	Professor	W16	PhD	25
17	Special Physical Education	Senior researcher	W17	PhD	20
18	Physical Therapy	Professor	W18	PhD	15
19	Physical Therapy	Professor	W19	PhD	20
20	Physical Therapy	Professor	W20	PhD	40

**Table 2 healthcare-11-03031-t002:** Survey results for the determination of evaluation factors.

		Evaluation Factors	Mean	SD	I-CVI	CV
Physical function	Health-related factors	Muscle strength	4.6	0.49	1	0.11
		Muscular endurance	4.3	0.56	0.95	0.13
		Cardiovascular endurance	4.1	0.62	0.85	0.15
		Flexibility	4.35	0.73	0.85	0.17
		Body composition	4.1	0.70	0.8	0.17
	Skill-related factors	Static balance	4.6	0.49	1	0.11
		Dynamic balance	4.8	0.40	1	0.08
		Coordination	4.7	0.46	1	0.10
	ADL-related factors	Range of motion	4.55	0.50	1	0.11
		Upper & lower extremity function	4.75	0.43	1	0.09
		Muscle tone	4.3	0.71	0.85	0.17
		ADL	4.7	0.46	1	0.10
		IADL	4.35	0.57	0.95	0.13
Mental function		Cognition	4.5	0.67	0.9	0.15
		Depression	4.15	0.65	0.85	0.16
		Fear of falling	4.45	0.67	0.9	0.15
		Self -efficacy	4.45	0.74	0.85	0.17
		Body awareness	4.4	0.73	0.85	0.17
Social ability		Disability acceptance	4.4	0.58	0.95	0.13
		Communication skill	4.4	0.66	0.9	0.15
		Social participation	4.45	0.59	0.95	0.13
		Quality of life	4.65	0.57	0.95	0.12

ADL, activities of daily living; CV, positive coefficient of variation; IADL, instrumental activities of daily living; I-CVI, content validity of individual items; SD, standard deviation.

**Table 3 healthcare-11-03031-t003:** Survey results for the determination of evaluation factors.

Evaluation Factors	Assessment Tools	Mean		SD		I-CVI		CV	
		Importance	Feasibility	Importance	Feasibility	Importance	Feasibility	Importance	Feasibility
**I. Physical function: Health-related factors**									
1. Muscle strength	[M] **Manual muscle test**	4.42	4.26	0.67	0.64	0.85	0.85	0.15	0.15
	[M] Isometric muscle test	3.26	3.11	0.71	0.64	0.4	0.25	0.22	0.21
	[I] Dynamometer	3.53	3.47	0.68	0.68	0.6	0.55	0.19	0.20
	[I] Back muscle dynamometer	2.89	2.68	0.55	0.46	0.1	0	0.19	0.17
	[I] 1-repetition maximum	2.68	2.63	0.73	0.67	0.1	0.05	0.27	0.25
2. Muscular endurance	[M] Step test	3.32	3.21	0.73	0.69	0.45	0.35	0.22	0.22
	[M] Step-box test	2.84	2.53	0.49	0.50	0.05	0	0.17	0.20
	[M] 30 s chair stand test	3.79	3.53	0.69	0.60	0.7	0.55	0.18	0.17
3. Cardiovascular endurance	[M] 6 min walk test	3.68	3.58	0.73	0.75	0.6	0.5	0.20	0.21
	[M] Pacer test (20 m)	2.68	2.42	0.57	0.67	0	0	0.21	0.28
	[I] Treadmill test	2.79	2.53	0.77	0.68	0.15	0.05	0.27	0.27
	[I] Ergometer test	3.11	2.89	0.72	0.55	0.3	0.1	0.23	0.19
	[I] **FVC, FEV1**	4.05	4.00	0.51	0.46	0.85	0.85	0.13	0.11
	[I] Power breathe	3.79	3.74	0.52	0.55	0.7	0.65	0.14	0.15
4. Flexibility	[M] Sit-and-reach test	3.37	3.32	0.67	0.65	0.45	0.4	0.20	0.20
	[M] Trunk forward flexion test	2.84	2.63	0.81	0.74	0.2	0.1	0.29	0.28
	[M] Shoulder flexibility test	3.74	3.68	0.55	0.65	0.65	0.55	0.15	0.18
5. Body composition	[I] Skinfold technique	3.21	3.21	0.83	0.83	0.3	0.3	0.26	0.26
	[I] **Body composition analyzer**	4.11	4.21	0.55	0.61	0.85	0.85	0.13	0.15
	[M] Body mass index	3.79	3.89	0.61	0.64	0.65	0.7	0.16	0.16
**I** **. Physical function: Skill-related factors**									
6. Static balance	[M] **Functional reach test**	4.37	4.32	0.58	0.57	0.9	0.9	0.13	0.13
7. Dynamic balance	[M] **Timed up and go**	4.37	4.26	0.81	0.78	0.75	0.75	0.19	0.18
	[M] **Berg balance scale**	4.32	4.21	0.65	0.69	0.85	0.8	0.15	0.16
	[M] Tinetti test	3.68	3.63	0.73	0.67	0.6	0.5	0.20	0.18
	[M] Short physical performance battery	3.79	3.68	0.61	0.65	0.65	0.55	0.16	0.18
	[M] 4-square step test	3.21	3.21	0.61	0.61	0.3	0.3	0.19	0.19
8. Coordination	[M] **Trunk impairment scale**	4.21	4.05	0.52	0.51	0.9	0.85	0.12	0.13
	[I] Force plate measurements	4.00	3.84	0.56	0.67	0.8	0.75	0.14	0.17
	[M] **Postural assessment scale for stroke**	4.21	4.11	0.69	0.55	0.8	0.85	0.16	0.13
**I** **. Physical function: ADL-related factors**									
9. Range of motion	[I] **Goniometer**	4.58	4.47	0.49	0.50	0.95	0.95	0.11	0.11
10. Upper & lower extremity function	[M] **Motor assessment scale**	4.42	4.32	0.67	0.65	0.85	0.85	0.15	0.15
	[I] Jebsen—Taylor hand function test	4.32	3.89	0.57	0.64	0.9	0.7	0.13	0.16
	[M] **Manual function test**	4.26	4.21	0.55	0.52	0.9	0.9	0.13	0.12
	[I] Grooved pegboard test	3.63	3.47	0.67	0.68	0.5	0.35	0.18	0.20
	[I] Box and block test	4.05	4.00	0.76	0.79	0.7	0.65	0.19	0.20
	[M] Fugl—Meyer assessment	4.16	3.58	0.67	0.82	0.8	0.55	0.16	0.23
11. Muscle tone	[M] **Modified Ashworth scale**	4.53	4.21	0.50	0.69	0.95	0.9	0.11	0.16
12. ADL	[S] Functional independence measure	4.42	3.79	0.59	0.77	0.9	0.75	0.13	0.20
	[S] **Modified Bathel index**	4.32	4.00	0.57	0.73	0.9	0.8	0.13	0.18
13. IADL	[S] **K-IADL**	4.37	4.16	0.48	0.49	0.95	0.9	0.11	0.12
**Ⅱ** **. Mental function**									
14. Cognition	[S] **Mini-mental status examination**	4.47	4.32	0.50	0.57	0.95	0.9	0.11	0.13
	[S] Korean version of Montreal cognitive assessment	4.05	3.89	0.60	0.64	0.8	0.7	0.15	0.16
15. Depression	[S] Geriatric depression scale short form	3.74	3.63	0.71	0.67	0.55	0.5	0.19	0.18
	[S] **Patient health questionnaire**	4.32	4.11	0.65	0.55	0.85	0.85	0.15	0.13
	[S] Beck depression inventory	4.05	3.79	0.69	0.69	0.75	0.6	0.17	0.18
16. Self-efficacy	[S] **Self efficacy questionnaire**	4.53	4.26	0.50	0.44	0.95	0.95	0.11	0.10
17. Fear of fall	[S] **Fear of falling questionnaire**	4.42	4.26	0.59	0.55	0.9	0.9	0.13	0.13
	[S] Falls efficacy scale—International	4.21	3.95	0.41	0.39	0.95	0.85	0.10	0.10
18. Body awareness	[S] **Body awareness** **questionnaire**	4.42	4.05	0.49	0.60	0.95	0.9	0.11	0.15
**Ⅲ** **. Social ability**									
19. Disability acceptance	[S] **Disability acceptance scale**	4.58	4.32	0.49	0.46	0.95	0.95	0.11	0.11
20. Communication skill	[S] **Holden communication scale**	4.47	4.26	0.60	0.55	0.9	0.9	0.13	0.13
21. Social participation	[S] **Participation in social activities and environmental factors**	4.47	4.26	0.50	0.44	0.95	0.95	0.11	0.10
22. Quality of life	[S] **EQ-5D**	4.26	4.26	0.78	0.64	0.75	0.85	0.18	0.15
	[S] **WHO quality of life**	4.42	4.21	0.49	0.41	0.95	0.95	0.11	0.10

Note: Assessment tools that satisfied the stop criteria (mean ≥ 4.0, I-CVI ≥ 0.75, CV < 0.50) were written in bold. [M] Manual: assessment tools not required. [I] Instrument: assessment tool that requires an instrument. [S] Survey: assessment tool that requires questionnaires. EQ-5D, European quality-of-life-5-dimensions questionnaire; FVC, forced vital capacity; FEV1, forced expiratory volume in one second; WHO, World Health Organization.

## Data Availability

The data presented in this study are available in the article.

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
