# Peer review of "Development of a Set of Assessment Tools for Health Professionals to Design a Tailored Rehabilitation Exercise and Sports Program for People with Stroke in South Korea: A Delphi Study"

_healthcare, 2023, doi:10.3390/healthcare11233031_

Round 1

Reviewer 1 Report

Comments and Suggestions for Authors

INTRODUCTION

1.         The introduction is quite good, I congratulate the authors. I think it would be important to talk about the biopsychosocial approach in the fourth paragraph.

2.         It is important in the final part of the introduction, to determine in the objective that a Delphi study is to be carried out.

METHODS

1.         The explanation of the Delphi method is good, but a little bit confusing, I recommend to review and rewrite some parts that are a little bit repetitive and confusing.

RESULTS

1.         In the tables the font size is different and there are some titles underlined and others not, I recommend to correct this.

DISCUSSION

1. It would be interesting to give an explanation of why none of the following items were appropriate (muscular endurance, flexibility and dynamic balance) and to discuss with other studies.

2. There are only two references in the whole discussion, to give consistency as the name of the section indicates, it would be necessary to discuss and contrast the information with other articles.

3.         I believe that more limitations could be included in your study.

REFERENCES

1.         It is necessary to review the references since not all of them are in the same format.

Reviewer 2 Report

Comments and Suggestions for Authors

I think it is wonderful that authors have constructed manuscripts based on many references.

However, I have the following questions.

The "Rehabilitation Exercise and Sports" (RES) program developed by the authors is an assessment tool from 6 months to 3 years after stroke onset. Stroke rehabilitation needs to start in the acute phase. Given the recurrence rate of stroke, why are there so many kinesiology specialists and no neurosurgeons on the expert panel?

Thank you very much.
